# Approach to Contemporary Risk Assessment, Prevention and Management of Thrombotic Complications in Multiple Myeloma

**DOI:** 10.3390/cancers14246216

**Published:** 2022-12-16

**Authors:** Despina Fotiou, Meletios Athanasios Dimopoulos, Efstathios Kastritis

**Affiliations:** Department of Clinical Therapeutics, School of Medicine, National and Kapodistrian University of Athens, 11528 Athens, Greece

**Keywords:** multiple myeloma, thromboembolism, thromboprophylaxis, VTE risk assessment scores

## Abstract

**Simple Summary:**

There is a significant risk of thrombotic complications in patients with multiple myeloma. Patient-related, disease-related, and treatment-related risk factors contribute to the observed prothrombotic environment. The prevention of thrombosis requires implementing risk assessment tools and pharmacological thromboprophylaxis. Newer oral anticoagulants are increasingly being used in this setting, but robust data emerging from controlled clinical trials are lacking. Despite the use of multiple myeloma-specific risk-assessment tools and the implementation of thromboprophylaxis based on current guidelines, the risk of thromboembolism remains significant, pointing to the need to understand this common complication further.

**Abstract:**

Multiple myeloma (MM) is associated with an increased risk of thrombotic complications, which remains substantial despite the implementation of thromboprophylaxis. The procoagulant state that characterizes the disease is multifactorial, and a greater understanding of the underlying pathophysiology is required to inform appropriate thrombosis prevention. Currently, there is a shift towards using direct oral anticoagulants (DOACs) in this setting; head-to-head comparisons in the context of controlled clinical trials between class agents are still missing. MM-specific VTE risk assessment scores have been developed to optimize management and minimize the associated mortality/morbidity. Their clinical utility remains to be evaluated. The value of adding biomarkers to clinical scores to optimize their performance and increase their discriminatory power is also under assessment.

## 1. Introduction

Multiple myeloma is characterized by a procoagulant predisposition, and patients with multiple myeloma (MM) have an approximately nine-fold higher risk for venous thromboembolism (VTE) than the general population [1,2]. The emergence of VTE as a significant complication in MM patients parallels the use of multiagent chemotherapy regimens (doxorubicin, vincristine, and more), and later on, the introduction of immunomodulatory agents (IMiDs). The association between IMiDs and VTE risk is well established in the literature [1,2]. Both venous and arterial thrombosis are a leading cause of morbidity, but evidence of the association between VTE and mortality in MM is limited and conflicting [3,4,5,6]. As the therapeutic options for the disease expand, and MM increasingly acquires a “chronic” disease trajectory, the question of optimal VTE prophylaxis becomes increasingly relevant. The International Myeloma Working Group (IMWG) [7], the European Myeloma Network (EMN) and the National Comprehensive Cancer Network (NCCN) [8] have published guidance on the prevention of MM-associated thrombosis. A consensus-based position paper was also released earlier this year [9]. Appropriate risk identification for VTE is advocated for optimal care of patients with MM, and central to the framework of international guidelines is the ability to appropriately risk stratify MM patients for VTE. Patients with myeloma should be risk-stratified for VTE risk and then offered aspirin; low molecular weight heparin (LMWH); or according to more recent updates of guidelines, direct oral anticoagulants (DOACs), for thromboprophylaxis. Three MM-specific VTE risk assessment models have been developed and validated; one by International Myeloma Working Group (IMWG) [7], and more recently, the SAVED and the IMPEDE VTE [10] models. Recognizing and understanding the multiparametric nature of the procoagulant environment observed in MM is paramount for optimal risk stratification. Interestingly, VTE rates are also higher in the pre-symptomatic stages of the disease, such as monoclonal gammopathy of undetermined significance (MGUS) [11,12,13,14]. Unfortunately, little progress has been made in the field over the last few years, and VTE rates remain high, often over 10%, both in the clinical trials and real-world settings [1,2]. Data supporting the safety and efficacy of DOACs are increasingly emerging, but head-to-head comparisons between agents are still missing, and randomized control trial (RCT) data are scarce [15,16,17,18,19]. The type and duration of thromboprophylaxis remain controversial. The optimal prevention strategy, therefore, remains to this date an unmet clinical need. Risk-assessment algorithms have to be revised to reflect more accurately prothrombotic risk. Following that, guideline implementation in everyday clinical practice must be ensured.

In this review, we present current approaches to the management and prevention of thrombotic complications in patients with MM, and we highlight the necessity better to characterize the unique coagulation profile of the MM patient.

## 2. Literature Review

### 2.1. Risk Factors for Thrombosis

Thrombogenicity in MM is multiparametric. Patient-specific “standard” risk factors, features related to disease biology and treatment-related factors interact to formulate the thrombotic risk observed in each patient (see Figure 1).

#### 2.1.1. Patient-Related Risk Factors

Standard risk factors for thrombotic risk apply for the MM patient [20,21]. The incidences of factor V Leiden and PTG20210A polymorphisms in patients with MM are similar to the general population, and no other genetic variants have been described [22,23].

#### 2.1.2. Disease-Specific Risk Factors

A new diagnosis of multiple myeloma is considered a risk factor for thrombosis compared to relapsed/refractory disease [2]; however, VTE rates remain high also in the relapsed setting. As reported recently in a meta-analysis, the VTE incidence per 100 patient-cycles was comparable in newly diagnosed patients and in the relapsed setting, at 1.2 and 1.2, respectively [24]. Numerous studies have confirmed that most events occur within six months of treatment initiation, and this finding is independent of the treatment type [1,4]. Several groups have demonstrated that patients with MGUS and smoldering MM are also at a higher risk of developing VTE compared to the general population [11,12,13,14], in support of the notion that MM as a disease is a hypercoagulable state. In a large hospital-based study by Kristinsson et al. [11], during 17 years of follow-up, a 3.3-fold increase in DVT risk was observed in MGUS patients (n = 2374) compared to the healthy population. More recently, another large population-based Danish cohort study reported an incidence rate of 4.0 VTEs/1000 person-years in MGUS patients, and the IRR for VTE compared to a healthy control cohort matched with MGUS patients was 1.37 [25].

The exact mechanisms that underly disease-related hypercoagulability in MM remain to be delineated. There are data to support that hyperviscosity, inhibition of natural anticoagulants [26] and an increase in procoagulant inflammatory cytokines such as interleukin-6 (IL-6) and tumor necrosis factor-a (TNFa) all contribute to the prothrombotic environment observed. Monoclonal paraprotein also interferes with fibrin polymerization [27], clot retraction and clot lysis time [28,29], and hypofibrinolysis is observed [30]. More recently, immunoglobulin k light chain was found to be associated with dysfibrinogenemia through a mechanism of fibrin interference [31]. Chalayer et al. recently identified serum gamma globulin levels >27 g/L (2.8 [1.2–6.8], *p* = 0.02) as an independent prognostic factor for VTE occurrence in patients eligible for autologous transplantation who receive bortezomib, dexamethasone and lenalidomide induction.

Other proposed mechanisms include acquired activated protein C resistance; lupus antibody anticoagulant activity by the monoclonal component; increased microparticle-associated tissue factor (TF); elevated von Willebrand factor (vWF), fibrinogen or factor VIII; decreased protein C; and increased thrombin generation [26,27,32,33,34]. A recent review summarized the data regarding the association between increased vWF/FVIII levels in patients with MM, increased risk of thrombosis and decreased overall survival [35]. Altered platelet function has also been reported, but data are not conclusive yet; Egan et al. [36] reported decreased platelet aggregation in MM patients but no link between platelet hyporeactivity and paraprotein levels, and Sullivvan et al. reported platelet hyperactivation with increased CD63, PAC-1 expression and annexin V in resting platelets of MM and MGUS patients compared to healthy controls [37]. A recent study assessed thrombin generation, procoagulant phospholipids (PPL), neutrophil extracellular traps (NETs) and circulating microvesicle (MV)-associated TF in 38 patients and 19 MGUS patients, and compared them to healthy controls [14]. As confirmed by other studies [38,39,40], MM and MGUS patients have increased thrombin generation and PPL activity compared to healthy controls. Cell-free DNA (cfDNA) is not a specific marker of NETs and increased thrombotic risk in MM patients, but it may reflect NETs formation [41,42,43]. Recent data demonstrated that activation of PAD4 might induce NETs release directly in myeloma cell lines [44]. Further studies must determine whether these findings can be attributed to the procoagulant NETs activity. It should be noted that no patients in the study developed VTE, so a correlation between these findings and VTE occurrence cannot be demonstrated.

#### 2.1.3. Treatment-Related Risk Factors

Treatment-related factors are key determinants of the final thrombotic risk of MM patients (Table 1). The introduction of IMiDs was associated with a significant rise in VTE occurrence among the MM population. Thalidomide or lenalidomide (Len) monotherapy baseline risk is around 3–4%, but this increases substantially when combined with dexamethasone. The dose of dexamethasone and the partner drugs affect the risk significantly; the risk increases up to 26% with the addition of high-dose dexamethasone or multiagent chemotherapy [45,46,47,48]. Lenalidomide maintenance also confers a significant thrombotic risk; both venous and arterial events were significantly higher compared to placebo in the myeloma XI trial [49]. Notably, the VTE risk persists following drug cessation [50]. Fewer data exist regarding pomalidomide; the VTE risk reported is around 2–3% when pomalidomide is administered in combination with dexamethasone. The VTE risk is therefore lower compared to lenalidomide-based combinations. It should be, however, taken into account that data come from the relapsed setting only and following the implementation of regular thromboprophylaxis in this patient population [51,52,53].

The exact mechanisms underlying the IMiD-induced thrombogenic effect are unknown, but higher levels of P-selectin, fibrinogen and homocysteine following lenalidomide treatment have been reported. Platelet activation is enhanced, and resistance to protein C, mediated by cytokines following IMiD-based therapy, has been reported [74]. High-dose dexamethasone increases the levels of P-selectin, vWF and FVIII [75], and there are reports of higher plasma thrombin levels following therapy with doxorubicin [76].

The proteasome inhibitor (PI) bortezomib is associated with a low risk of VTE [58,59]. A protective effect has even been reported in some studies when bortezomib is combined with IMiDs [55,60], but more recent data are conflicting; two RCTs did not show a significant difference in the all-grade risk of thrombosis when IMiD plus dexamethasone is combined with bortezomib [61,62]. In contrast to bortezomib, the second-generation PI carfilzomib has a different adverse effect profile and is associated with significant cardiovascular side effects, but less is known about its VTE risk. In the newly diagnosed setting, VTE rates reported for carfilzomib-IMiD-based combinations range from 5 to 14%, but a uniform approach to thromboprophylaxis is missing [63,67,68]. In a recent study, the VTE rate was 16.1% for patients on KRD (carfilzomib, lenalidomide, dexamethasone), and it was 4.8% for patients on VRD (bortezomib–Rd), confirming the different profiles of the two PIs in terms of thrombotic risk [68]. The cause of carfilzomib-induced cardiotoxicity is suspected to be a result of endothelial dysfunction [77]. Whether the mechanisms of endothelial dysfunction are involved in venous thromboembolic events is still unknown, but many groups support that patients on the KRd combination should be considered high risk in terms of VTE risk [68].

Other agents with anti-myeloma activity, such as the monoclonal antibodies, daratumumab, belantamab and elotuzumab, and the oral proteasome inhibitor ixazomib, have not been associated with an increased risk of thrombotic complications [67,68,69,70], at least in the clinical trial setting. In the GRIFFIN study, daratumumab with VRD (DVRd) was compared to VRd, and the all-grade cumulative rates of VTE were 10% and 15%, respectively [73]. A post-hoc pooled analysis of the phase III CASTOR, POLLUX and MAIA trials demonstrated that daratumumab treatment did not affect VTE risk [78].

### 2.2. Risk Assessment Models for VTE in Multiple Myeloma

The importance of risk assessment models (RAMs) for thrombosis prediction in cancer patients has become well-established since the Khorana risk score was developed in 2008 [79]. However, its discriminative power for VTE risk in patients with MM is poor, as reported by Sanfilippo et al. (assessment in 2870 MM patients) [80]. In the MM population, the only parameter that retains its predictive significance for VTE is the white blood cell count [81]. Research efforts focused on the development of MM-specific RAM. Three MM-specific RAMs have been published and validated to date for predicting VTE in MM patients: the earlier International Myeloma Working Group (IMWG) [7], and more recently, the SAVED [82] and the IMPEDE VTE [10] models. The three scores are presented in detail in Table 2. The NCCN guidelines historically adopted the IMWG score, but it has performed poorly when applied in the clinical trial setting (Table 2). Bradbury et al. reported the thrombosis rate in Myeloma XI and XI clinical trials and demonstrated the suboptimal predictive power of the IMWG model [49]. Half of the patients who developed VTE in the Myeloma XI trial were not classified as high risk using the IMWG scoring system [55]. A minimum of 3 months of treatment with LMWH was recommended for high-risk patients, and aspirin for low-risk patients. Out of the 3838 patients treated in the trial, the VTE rate was 11.8%. At the time of the event, 87.7% were on thromboprophylaxis. The use of the IMWG score outside clinical trials has been moderate [83]. In a retrospective review, 80 MM patients with VTE were identified and matched to controls (MM patients without VTE); among patients with VTE, 82% were considered high risk and 18% low risk at baseline. Only 19% of patients received appropriate thromboprophylaxis (LMWH, aspirin, none) based on IMWG risk-score baseline assessment [83].

The IMPEDE-VTE score was developed using retrospective data from the Veterans Administration Central Cancer Registry (VACCR) in 4446 patients, and it was subsequently validated using the Surveillance, Epidemiology, End Results (SEER) Medicare database. The derivation cohort included patients who were diagnosed with MM between 1 September 1999 and 30 June 2014 and had received chemotherapy for MM; patients were followed up retrospectively for 180 days to determine the occurrence of VTE events. The validation cohort from the SEER database included patients who were Medicare eligible (≥65 years old) and were diagnosed with MM from January 2007 to December 2013. Three risk groups were identified; the respective 6-month cumulative incidence of VTE following treatment initiation was 3.3% for the low-risk group, 8.3% for the intermediate-risk group and 15.2% for the high-risk group [10] (Table 2). A recent validation of the IMPEDE score in 575 patients with NDMM reports a 6-month cumulative incidence for VTE of 5% (95% CI: 2.1–7.9) in the low-risk group, compared to 12.6% (95% CI: 8.9–16.4%) and 24.1% (95% CI: 12.2–36.1) in the intermediate- and high-risk groups, respectively (*p* < 0.001 for both) [84]. The SAVED risk score (surgery, Asian race, VTE history, eighty years old, dexamethasone) was developed around the same time using the SEER database and was validated using the Veterans cohort [82]. It was developed only in patients on IMiD-based therapy; patients were stratified as low or high risk for VTE with hazard ratios of 1.85 (*p* < 0.01) and 1.98 (*p* < 0.01) in the derivation and validation cohorts, respectively. (Table 2) In the recent retrospective study by Piedra et al., who reported VTE rates for KRd and VRd and compared rivaroxaban to ASA, the SAVED model did not predict VTE, but due to the retrospective design of the study high-risk patients on anticoagulation were excluded [68]. The SAVED scores were calculated retrospectively in the GRIFFIN study for patients in both the Dara-VRd and VRd arms. Median SAVED scores in both groups were in the low-risk category (0 and 0.5 scores, respectively); the VTE risk was low, but the incidence of VTE in both groups was relatively high [73]. In addition, only 60% of patients in the Dara-RVd and 67% in the RVd arm were on antithrombotic treatment at the time of VTE [73]. The performances of the SAVED and IMPEDE-VTE risk scores were retrospectively assessed in the CASTOR, POLLUX and MAIA studies; both scores had good discrimination for patients on non-daratumumab-based regimens, and the SAVED score performed better in patients receiving daratumumab [78].

Validation analysis of the three scores was performed using the two large MM cohorts (SEER-Medicare and Veterans Administration Healthcare System); Harrell’s c-statistics were 0.52 and 0.55, respectively, for the IMWG model, and slightly better for the newer models. Harrell’s c-statistic was 0.60 for the IMPEDE-VTE model and 0.64 for the SAVED model, respectively [85]. Based on these findings, the IMPEDE-VTE and the SAVED models are preferentially recommended for utilization in the latest guidelines [86]. However, as Stefano et al. argue in their latest consensus paper [9], there are insufficient data to recommend one specific RAM over the other in clinical practice. We should also note that both IMPEDE-VTE and SAVED scores were developed using patients who received treatment mostly before the era of novel agent availability used currently in clinical practice.

As summarized in a previous review by our group [87], an expanding body of literature has identified several serologic biomarkers that may help predict VTE risk in MM. Among measurable coagulation parameters, D-dimer levels and thrombin generation have been studied as potential biomarkers that could optimize the predictive power of risk-assessment VTE models [82,83]. A recent study examined the performance of the IMPEDE VTE score when combined with D-dimer levels in newly diagnosed MM patients; patients with D-dimer levels in the highest quartile had a 2-fold increase in the risk of VTE after adjusting for IMPEDE VTE score ((hazard ratio (HR) 2.04)) [88].

### 2.3. Primary VTE Prevention in Multiple Myeloma

Clinical trial data regarding pharmacological thromboprophylaxis in patients with MM are limited. We identified only four randomized clinical trials in the literature that compare antithrombotic agents for VTE prevention for patients with MM on IMiDs [89], and further evidence comes from non-randomized observational trials. ASA was compared to fixed low-dose warfarin (1.25 mg/day) and to LMWH (enoxaparin 40 mg/day) in 667 NDMM patients who received thalidomide in a randomized clinical trial (RCT). No significant differences were demonstrated between the three agents; the VTE rate at six months was 6.3% in the ASA group, 8.2% in the warfarin group and 5% in the LMWH group [18]. The rates of cardiovascular events, all-cause mortality and bleeding events also did not differ between groups. Aspirin (100 mg/day) was also compared to enoxaparin 40 mg/day in another RCT in MM patients who received lenalidomide, and no significant differences were reported; the VTE rate was 2.2% in the aspirin group and 1.2% in the enoxaparin group [19]. Pooled data analysis in a Cochrane systematic analysis demonstrated no superiority or inferiority regarding ASA use compared to LMWH in this setting, and appraisal resulted in very low-certainty evidence [89]. It should be noted, however, that high-risk patients for thrombotic complications were excluded from both trials.

The VTE rate was 10.7% for patients on ASA and 1.4% for patients on prophylactic LMWH in a retrospective review of 1126 patients with MM, questioning its protective role even in low-risk patients [90]. A meta-analysis published in 2018 in 1964 patients included data from retrospective and longitudinal studies also; ASA was found to be superior to no intervention (OR = 0.20; 95%CI: 0.07–0.61, *p* = 0.005; I^2^ = 41%) and inferior to LMWH (OR = 2.60; 95%CI: 1.08–6.25; *p* = 0.03; I^2^ = 0%) [91]. In another large retrospective review (4892 patients with MM), 586 patients had a VTE event. After adjusting for IMiD use and prior VTE history, aspirin did not reduce the risk of VTE enough to justify its role [92]. ASA thromboprophylaxis is, therefore, not recommended during the first six months of first-line treatment in MM patients who receive IMiDs [93]. In the Myeloma XI study, patients were risk-stratified at baseline; among patients who experienced a VTE, 9.2% were on a therapeutic dose of warfarin, 44.1% on prophylactic LMWH dose and 31% on aspirin. A direct comparison is not possible, given the baseline risk stratification [55].

DOACs have become the most favored class of drugs in the management and prevention of thromboembolic complications. The benefits of apixaban and rivaroxaban in the setting of primary VTE prevention in intermediate- to high-risk ambulatory cancer patients was demonstrated in the randomized controlled trials AVERT and CASSINI [94,95]. Very few MM patients were enrolled in these studies, making generalization of the evidence to the MM setting challenging. Only 2.6% of patients in the AVERT study had MM, and myeloma was an exclusion criterion in the CASSINI trial. Data to support the use of DOACS in primary VTE prevention for MM patients has, however, more recently accumulated from smaller studies. In a cohort of 305 NDMM patients, rivaroxaban (10 mg once a day) thromboprophylaxis in patients who received KRd had superior efficacy compared to ASA (81 mg), with VTE rates being 16% and 5%, respectively. In addition, no increase in bleeding risk was reported [68]. The safety and efficacy of apixaban thromboprophylaxis at 2.5 mg twice daily for at least six months in patients with NDMM who receive IMiDs have been evaluated in four recent trials in a total of 306 patients [54,56,57,69]. In the pooled data, two VTE events (0.6%) were reported, along with three episodes of major hemorrhage (1%). These results were not validated in comparison to another method for VTE prevention in a randomized trial, but are encouraging. One group compared the rate of thromboembolic complications before and after using apixaban 2.5 mg twice daily as routine thromboprophylaxis for patients on IMiDs. Before the policy change, the VTE rate was 20.7% in patients on ASA and 7.4% in patients who received prophylactic-dose LMWH, whereas there were no VTE events after 2014 and the change in practice, within six months of treatment initiation [56].

The type, intensity and duration of thromboprophylaxis should be patient-specific and tailored to each patient’s baseline thrombotic and hemorrhagic risk [9]. Important issues when considering the mode of thromboprophylaxis include, of course, effectiveness, but also the route of administration, patient convenience—one should opt for the least intrusive method and safety in the context of frailty—age, cyclical thrombocytopenias, minimizing bleeding risk and impaired renal function. Patient preference, probability of compliance with treatment and cost also need to be considered. Drug–drug interactions and the use of strong cytochrome P inducers/inhibitors should be taken into account. Among anti-myeloma agents, dexamethasone is a strong inducer of CYP3A4 and P-glycoprotein and an inhibitor of P-glycoprotein; bortezomib is a CYP3A4 inhibitor; and venetoclax is an inhibitor of P-glycoprotein [96,97]. The initial choice of thromboprophylaxis should be adapted to renal function. A patient with thrombocytopenia is another challenge, as clear-cut instructions and thresholds for using different agents are absent [96,97,98]. Other factors to consider are extremes of body weight, history of heparin-induced thrombocytopenia (HIT), availability of reversal agents and single/dual antiplatelet agent use.

All patients with MM starting anti-myeloma therapy should be assessed for thrombotic risk and should be considered for thromboprophylaxis. IMWG [7] and EMN [99] guidelines in 2014 and 2015 recommend using the IMWG score in patients with MM who receive IMiDs. Based on the baseline risk stratification, it is recommended that low-risk patients (no risk factors or one risk factor only, which is not treatment-related) should receive 100 mg aspirin for VTE prophylaxis. Otherwise, prophylactic LMWH or warfarin (with target INR 2-3) should be used. Based on the IMWG and EMN recommendations, thromboprophylaxis should be continued for at least four months with an option to switch to aspirin after that. The recent 2021 NCCN guidelines [8,100] have incorporated the IMPEDE VTE and SAVED scores into baseline risk assessment. Low-risk NDMM patients (≤3 points IMPEDE VTE score or <2 by SAVED score) should receive aspirin at 81–325 mg, and high-risk patients (≥4 points by IMPEDE VTE score or ≥2 SAVED score) should be offered LMWH (enoxaparin 40 mg or equivalent). Other options for these patients include rivaroxaban 10 mg daily, apixaban 2.5 mg twice daily, fondaparinux 2.5 mg daily and warfarin (target INR 2-3). In a recent consensus paper, Stefano et al. [9] recommend that patients who are low risk for thrombosis should receive no thromboprophylaxis or low-dose aspirin. According to a consensus paper, prophylactic-dose LMWH should be the first choice for all other patients. Stefano et al. argue that there is no substantial evidence to support the use of DOACs over LMWH, even though preliminary data on the efficacy and safety of apixaban and rivaroxaban are promising. Thromboprophylaxis should be considered a process that requires continuous revaluation, as the thrombotic and bleeding risks often change throughout the course of the disease of MM patients. Prophylaxis should continue as long as the thrombotic risk is present and no contraindication exists.

### 2.4. Treatment of VTE and Secondary Prevention of VTE

The treatment of acute VTE in cancer patients is well-established [101]. The same principles apply to the treatment and secondary prevention of thromboembolic events in patients with MM. Management should also be disease-specific and patient-tailored. In cancer patients, there is a gradual transition from the use of LMWH to the use of DOACs, as data regarding the safety and efficacy of these agents are accumulating [95,102,103,104]. It should be noted that hematological malignancies represent a small proportion of patients enrolled in these studies. A meta-analysis in 2894 cancer patients demonstrated superiority for DOACs in efficacy, as VTE recurrence was lower, at 5.2% compared to 8.2% for LMWH. There was, however, a non-significant increase in major bleeding events and a significant increase in non-major bleeding events [105]. Based on current guidelines, when a VTE event occurs in patients with MM, the bleeding risk should be assessed, and patients should receive anticoagulation for at least six months [86]. De Stefano et al.’s recently published consensus paper summarizes recommendations for treating acute thrombotic events and secondary antithrombotic prophylaxis [9]. In the case of ongoing treatment with IMiDs, indefinite thromboprophylaxis should be considered. The long-term thromboprophylaxis should also be considered in the relapsed or high-disease burden setting until a response is achieved. The authors suggest that with regard to thrombocytopenia, the same principles apply as for primary thromboprophylaxis. Dose-adjusted LMWH or VKA is most appropriate for patients with severe renal insufficiency. DOACs can be considered for patients with mild or moderate renal insufficiency based on each drug’s summary of product characteristics.

## 3. Arterial Thromboembolic Risk in MM

The arterial thromboembolic risk in patients with MM is less well characterized and understood. MM is a disease of older adults; several risk factors for arterial thromboembolic events (ATEs), coronary artery disease and cerebrovascular disease are shared, as the burden of cardiovascular comorbidities increases in the older patient population.

Before the introduction of newer IMiDs and PIs, several population studies reported an increased risk of ATE in MM patients compared to matched controls; Kristinsson et al. reported a 3.8% rate of ATE in MM patients [11]. In a prospective cohort of MM patients who received doxorubicin–dexamethasone-based regimens, the ATE rate was 5.6%, and thalidomide therapy (in combination with doxorubicin and dexamethasone) was not a risk factor for ATE [106].

In the Myeloma XI rial, ATE incidence was lower, at 1.3% and 2.4% in transplant-eligible and ineligible patients, respectively [49]. In a cohort of 934 consecutive MM patients treated at the Cleveland Clinic, 25 ATE events were observed within a year from treatment initiation; the cumulative incidence of ATE at six months was 2%, and at 12 months, 2.7% [107]. There was no significant association between the use of different thromboprophylaxis agents and the incidence of ATE. The use of IMiDs in induction was also not associated with an increase in ATE risk (HR = 0.54, 95% CI, 024–1.20, *p* = 0.13). The only disease-specific risk factor for ATE occurrence was ISS stage III disease. ATE occurrence is associated with worse OS, but a causal relationship is not easy to establish due to the co-existence of other comorbidities and higher tumor burden in most cases. The association between IMiD use and ATE risk is less well-established compared to VTE risk and IMiD use. The event rate is low to allow observation of meaningful differences in ATE risk in the context of clinical trials [108]. IMiD maintenance in the Myeloma XI trial was associated with a slight increase in ATE risk [49]. According to the drug’s summary of product characteristics (SPC) and data from two phase-3 RCTs in 704 patients with MM, the incidences of myocardial infarction and cerebrovascular events were 1.98% and 3.4%, respectively, for patients in the lenalidomide–dexamethasone arm (and 0.57% and 1.7%, respectively, in patients treated with dexamethasone alone).

The risk of ATE In MM is mainly linked to modifiable risk factors, such as smoking, diabetes, hyperlipidemia and a pre-existing history of arterial thromboembolic events. All patients should undergo a thorough risk assessment for ATE prior to treatment initiation. There is a paucity of data with regard to primary prophylaxis for ATE in patients with MM. Ongoing trials in cancer patients will hopefully provide data to guide future practice. A phase III trial compared nadroparin (an LMWH) to a placebo for VTE and ATE prevention in patients with solid tumors and reported a 50% decrease in ATE risk in the nadroparin arm [109]. The CASSINI trial also demonstrated a lower incidence of ATE in patients who received rivaroxaban compared to placebo (1.0% vs. 1.7%), which was not statistically different [110].

## 4. Discussion

The main challenges in the current management of thromboembolic complications in patients with MM are (1) the suboptimal performance of the risk stratification tools that are currently available and (2) the fact that the use of thromboprophylaxis is not routinely incorporated for all patients with MM in every day clinical practice. In the clinical trial setting, despite adequate thromboprophylaxis, patients on lenalidomide-based regimens still have a substantial risk of VTE of around 6%, indicating either that risk assessment models have poor discriminatory power or that the use of thromboprophylaxis is not optimal. There are also data to support that in the real-world setting; patients often receive inappropriate thromboprophylaxis based on baseline risk stratification and that the approach is usually not guideline based.

The SAVED and IMPEDE-VTE risk assessment models have improved discriminatory performance compared to the IMWG model and are thus currently the best available tools for VTE risk assessment. Their performance is, however, suboptimal, and better tools need to be developed to inform effective thromboprophylaxis for MM patients. As research efforts focus on furthering the understanding of the mechanisms that contribute to the prothrombotic environment in MM, adapted, risk-assessment tools are expected to perform better. Several groups have attempted to identify a generic coagulation biomarker that can accurately reflect VTE risk in MM patients. The complicated nature of this task reflects the complexity and heterogeneity of the coagulation profiles of MM patients. The combination of clinical parameters with biomarkers predictive of VTE risk in newer risk-assessment models is expected to increase their discriminative power and improve their sensitivity. Existing scientific data do not allow evidence-based recommendations for the use of DOACs over the standard-of-care LMWH. Emerging data are promising, but head-to-head comparisons are lacking, and we need to move away from a non-uniform, opinion-based approach to thromboprophylaxis.

To summarize current recommendations, all NDMM should have a baseline risk assessment concerning thrombotic complications using available tools. The baseline VTE risk should be weighed against the hemorrhagic risk in a patient-specific manner. The final choice of the agent used for thromboprophylaxis should be tailored to each patient. Finally, the baseline risk should be reevaluated throughout the disease course, as it is an evolving process, and pharmacological thromboprophylaxis should be adapted accordingly. Disease status (relapse or remission), performance status, patient mobilization, platelet count and renal function are a few of the factors that need to be reassessed at regular intervals. Aspirin should be reserved for very-low-risk patients. For other patients, given that there is no contraindication, a choice should be made between LMWH and DOACs. Figure 2 presents an algorithm for VTE prevention in MM patients.

## 5. Conclusions

Unfortunately, advances in the management of thromboembolic complications in patients with MM in recent years have not followed the evolution of the other anti-myeloma therapies, and thromboembolism remains a significant complication. To overcome the unmet need for thromboprophylaxis optimization, we need to address the gaps in existing knowledge. First, we need to understand better the procoagulant state observed in MM patients, the contribution of individual risk factors and the interplay among the disease-, patient- and treatment-specific prothrombotic mechanisms. Second, we need to identify biomarkers and develop tools that accurately reflect the thrombotic risk in MM patients. Incorporating coagulation biomarkers in the current risk assessment tools seems to be a potentially useful tool. Third, we need clinical trials that will provide robust data on the safety and efficacy of different modes of thromboprophylaxis, using risk-assessment tools to stratify patients. Finally, we need to increase the implementation of international guidelines in everyday clinical practice.

## Figures and Tables

**Figure 1 cancers-14-06216-f001:**
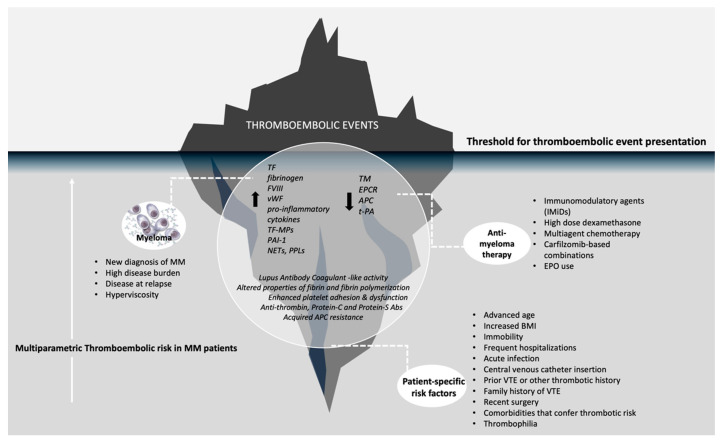
Risk factors for thrombosis in multiple myeloma. MM: multiple myeloma, TF: tissue factor, FVIII: factor VIII, vGF: von Willenbrand factor, TF-MPs: tissue factor-derived microparticles, PAI-1: plasminogen activator inhibitor, NETs: neutrophil extracellular traps, PPLs: procoagulant phospholipids, TM: thrombomodulin, EPCR: endothelial protein C receptor, APC: activated protein C, t-PA: tissue plasminogen activator, Abs: antibodies, EPO: recombinant erythropoein, BMI: body mass index, VTE: venous thromboembolism.

**Figure 2 cancers-14-06216-f002:**
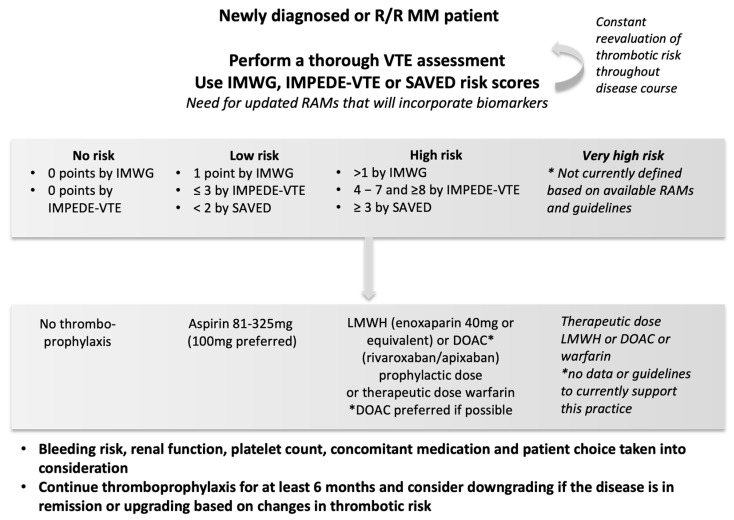
Algorithm for the assessment of VTE risk and choice of thromboprophylaxis. R/R: relapsed/refractory, MM: multiple myeloma, IMWG: international myeloma working group risk score, RAMs: risk assessment models, LWMH: low-molecular-weight heparin, DOAC: direct oral anticoagulant. * DOAC preferred if possible.

**Table 1 cancers-14-06216-t001:** Main clinical trial data regarding immunomodulatory agent (IMiD)-based regimens in the new treatment era.

	n		Regimen	Thromboprophylaxis	FU	VTE
Larocca et al., 2012 [19]	176 166	ND ND	Rd Rd	ASA 100 mg Enoxaparin 40 mg	6 m 6 m	8.2% 2.2%
Pegourie et al., 2019 [54]	104	RR/ND	Thal or Len	Apixaban 2.5 mg BID	6 m	1.9%
Fouquet et al., 2013 [46]	50	RR	Rd	All patients with event except 1 received aspirin, or LMWH or VKA	4 years	20%
Bradbury et al., 2017 [55]	3838	ND	CTD or CRD	87.7% of patients with event were on aspirin, LMWH, VKA	6 m	11.8%
Dimopoulos et al., 2014 [50]	353	RR	Rd	Not required per protocol	49 m	12.7%
Rajkumar et al., 2010 [47]	223 220	ND	R-D R-d	Thromboprophylaxis recommended and then mandatory	12.5 m	20% 9%
Storrar et al., 2019 [56]	70	ND	Len 21.5% Thal 78.5%	Apixaban 2.5 mg BID	6 m	0%
Cornell et al., 2020 [57]	50	All	Len 58% Pom 42%	Apixaban 2.5 mg BID	6 m	0%
Facon et al., 2018 [58]	1070	ND	Rdc/Rd18	NR	67 m	1.5%
Hou et al., 2013 [59]	199	RR	Rd	LMWH/warfarin for high-risk ASA for low-risk	15 m	0.5%
Hou et al., 2017 [60]	58 57	RR	Rd IRd	98% on thromboprophylaxis	20 m	0%
Lonial et al., 2015 [61]	318 317	RR	EloRd Rd	NR applied per institutional practice	24.5 m	6.6% 3.8%
Moreau et al., 2016 [62]	351 361	RR	Rd IRd	NR but almost all pts received	15 m	11% 8%
Stewart et al., 2015 [63]	389 392	RR	Rd KRd	NR	31.5 m	10.2% 6.2%
Richardson et al., 2014 [51]	108 221	RR	Pom PomDex	Aspirin unless contraindicated	14 m	3% 2%
Richardson et al., 2010 [64]	66	ND	VRD	At least aspirin in all patients	21 m	10%
Richardson 2014 [51,65]	64	RR	VRD	Aspirin, LMWH or VKA	44 m	1.6%
Durie et al., 2017 [66]	241 226		VRD Rd	Aspirin 325 mg	55 m	
San Miguel et al., 2013 [52]	300 150	RR	Pomd PomD	physicians’ discretion	10 m	2% 1%
Leleu et al., 2013 [53]	92	RR	Pd	70% aspirin, 40.5% LMWH, 14% VKA	23 m	4%
Kumar et al., 2020 [67]	527 526	ND	VRd KRd	NR	5 years	2.5% 5.7%
Piedra et al., 2022 [68]	124 99 82	ND	VRd KRd KRd	ASA 81 mg ASA 81 mg Rivaroxaban 10 mg OD	3 m	4.8% 16.1% 4.8%
Sayar et al., 2022 [69]	98 82	RR/ND	IMIDs	ASA 75 mg Apixaban 2.5 mg od	NR	4% 0%
Dimopoulos et al., 2016 [70]	285 281	RR	DRd Rd	ASA 77.7%, LMWH 24%, VKA 2.3%, DOAC 3%	13 m	All 5.5% *
Palumbo et al., 2016 [71]	243 237	RR	DVd Vd	ASA 21.5%, LMWH 10.6%, VKA 1.5% DOAC 2.4%	7 m	All 2.1% *
Facon et al., 2019 [72]	364 265	ND	DRd Rd	N/A data but LMWH or VKA required by protocol	28 m	All 12.4% *
Sborov et al., 2022 [73]	104 103	ND	D-VRd VRd	Per IMWG guidelines	38.6 m	10% 15.7%

VTE: venous thromboembolism, FU: follow up, ND: newly diagnosed, RR: relapsed/refractory disease, months, yr: years, NR: not reported, Rd: lenalidomide-dexamethasone, Len: lenalidomide, Thal: thalidomide: Pom: pomalidomide, ASA: aspirin, LMWH; low-molecular-weight heparin prophylactic dose, VKA: vitamin K agonists, CTD: cyclophosphamide-thalidomide-dexamethesone, CRD: cyclophosphamide–lenalidomide–dexamethasone, R-D: lenalidomide and high dose dexamethasone, R-d: lenalidomide and low-dose dexamethasone, Pomd: pomalidomide and low dose dexamethasone, PoMD: pomalidomide and high dose dexamethasone, VRd: bortezomib-lenalidomide- dexamethasone, KRd: carfilzomib-Rd, DRd: daratumumab-Rd, D-VRD: daratumumab-VRd, IMWG: international myeloma working group. * No difference in VTE rates in the three trials between Daratumumab containing arms and non-daratumumab containing arms, IMiD use was a risk factor.

**Table 2 cancers-14-06216-t002:** (a) IWMG score and algorithm for MM patient risk stratification. (b) IMPEDE and SAVED risk assessment models.

(a)		
IWMG Score and Algorithm for MM Patient Risk Stratification		
Patient-Related Risk Factors 1 Point for Each	Disease-Related Risk Factors: 1 Point for Each	Treatment-Related Risk Factors: Points as Seen Below:
Body mass index >25, age > 75, Personal or family history of VTE, Central venous catheter, Acute infection or Hospitalization, Blood clotting disorders or Thrombophilia, Immobility with a performance status of >1, Comorbidities (liver, renal impairment, Chronic obstructive pulmonary disorder, diabetes mellitus, chronic inflammatory bowel disease), Race (Caucasian)	Diagnosis of Multiple Myeloma Evidence of hyperviscosity	IMiD in combination with low-dose dexamethasone (<480 mg/month) (1 point) IMiD plus High-dose dexamethasone (>480 mg/month) or doxorubicin or multiagent chemotherapy (2 points) IMiD alone (1 point) Erythropoietin use (1 point)
Risk stratification and recommended thromboprophylaxis: 0 points: Low risk—None 1 point: Intermediate risk—Aspirin at 100 mg >1 point: High risk—Low molecular weight heparin at prophylactic dose or therapeutic dose of warfarin		
**(b)**		
**CLINICAL RAMs for VTE in MM**		
**IMPEDE VTE Score**		**SAVED * Score**
Immunomodulatory drug (+4) BMI ≥ 25 kg/m^2^ (+1) Pathologic fracture pelvis/femur (+4) Erythropoiesis-stimulating agent (+1) Dexamethasone (High-dose, ≥1600 mg/cycle) (+4) Dexamethasone Low-Dose (<160 mg/cycle) (+2) Doxorubicin (+3) Ethnicity/Race = Asian (−3) VTE history (+5) Tunnelled line/CVC (+2) Existing use of therapeutic warfarin or low molecular weight heparin (LWMH) (−5) Existing use of prophylactic LMWH or aspirin (−3)		Surgery (within the last 90 days) (+2) Asian Race (−3) VTE history (+3) Eighty (age ≥ 80 y) (+1) Dexamethasone dose Standard (120–160 mg/cycle) (+1) High (>160 mg/cycle) (+2)
Stratified risk groups based on a weighted scoring system		
Low risk (score of ≤3) Intermediate-risk (score of 4–7) High risk (score of ≥8)		High risk (score of ≥2) Low risk (≤1)

* for patients on IMiD-based regimens only.

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
