# Peer review of "Approach to Contemporary Risk Assessment, Prevention and Management of Thrombotic Complications in Multiple Myeloma"

_cancers, 2022, doi:10.3390/cancers14246216_

Round 1
Reviewer 1 Report
In this review, they present current approaches to management and prevention of thrombotic complications in patients with MM.
L 33 : Could you replace references 1 & 2 by recent references ?
L37 :Evidence of the association of VTE with mortality in MM is limited and conflicting.
5. Zangari M, Barlogie B, Cavallo F, Bolejack V, Fink L, Tricot G. Effect on survival of treatment‐associated venous thromboembolism in newly diagnosed multiple myeloma patients. Blood Coagul Fibrinolysis. 2007;18:595–98.
6. Zangari M, Tricot G, Polavaram L, Zhan F, Finlayson A, Knight R, et al. Survival effect of venous thromboembolism in patients with multiple myeloma treated with lenalidomide and high‐dose dexamethasone. J Clin Oncol. 2010;28:132–35.
L57 remain high, often over 10%, both in the clinical trial and real-world setting [17,18]
L60 why do you cite you here?
L89 : specify the incidence which let you tell « VTE rates remain high also in the relapsed setting, as reported 89 recently in a meta-analysis »
L93 : idem specify the l-incidence « higher risk of developing VTE compared to the general population [1,2,15,16] »
L96 : Could you explain the link between hypofibrinolysis and paraprotein ? « Hyperviscosity, inhibition of natural anticoagulants, an increase of procoagulant inflammatory cytokines like interleukin-6 (IL-6) and tumor necrosis factor-a (TNFa) and hypofibrinolysis (due to monoclonal paraprotein interference) all contribute to the thrombotic risk »
L 103 : all multiple myeloma treatment do not alter platelet function. Nowadays, anthracyclins are not used. It seems not justify this sentence « Altered platelet function has also been reported, which may explain the demonstrated efficacy of aspirin as thromboprophylaxis for low-risk MM patients »
L109-112 : which reference ? « MM patients also had increased levels of cell-free DNA (cfDNA) as a surrogate of NETs. Further studies must determine whether these findings can be attributed to the procoagulant NETs activity. It should be noted that no patients in the study developed VTE, so a correlation between these findings and VTE occurrence cannot be demonstrated.
L134 : Recent data are against. Please, add contradictory data : « The proteasome inhibitor (PI) bortezomib is associated with a low risk of VTE [50,51], 134 and even a protective effect has even been reported in some studies [47,52]. »
L 174 Define « appropriate prophylaxis » («Only 19% of patients received appropriate prophylaxis[66]. »)
In the table : please add the definition « Dexamethasone (High-dose) (+4) Dexamethasone Low-Dose »for impede vte score
What about gammaglobulines level data recently published?
L273 : Could you check the major hemorrhage incidence (1%) [81-84] in Storrar NPF, Mathur A, Johnson PRE, et al. Safety and efficacy of apixaban for routine thromboprophylaxis in myeloma patients 643 treated with thalidomide- and lenalidomide-containing regimens. Br J Haematol. 2019 Apr;185(1):142-144.
L 286 : platelet count >50000/mm3 ?
L 296 : Are you sure ? « Drug-drug interactions and the use of strong cytochrome P inducers/ inhibitors should be taken into account; fortunately, no anti-myeloma agents (except dexamethasone) is known to be a potent inhibitor or inducer of P-glycoprotein and P450 [86]. » See reference Sorigue 2020.
You referred NCCN 2022 as 10 instead of 69. NCCN Clinical Practice Guidelines in Oncology (NCCN Guidelines): Cancer-Associated Venous Thromboembolic Disease.: 487 National Comprehensive Cancer Network; July 22, 2016. [updated Version 1. 2016; cited 2018 December 12]. Available from: 488
Author Response
Reviewer 1:
In this review, they present current approaches to management and prevention of thrombotic complications in patients with MM.
- L 33 : Could you replace references 1 & 2 by recent references ?
Answer: Thank you for pointing this out. We have replaced references 1 and 2 with more recent references.
- L37 :Evidence of the association of VTE with mortality in MM is limited and conflicting.
Answer: Thank you for the comment we have changed the sentence to: “Both venous and arterial thrombosis are a leading cause of morbidity but evidence of the association between VTE and mortality in MM is limited and conflicting.” And we have added the two references.
- Zangari M, Barlogie B, Cavallo F, Bolejack V, Fink L, Tricot G. Effect on survival of treatment‐associated venous thromboembolism in newly diagnosed multiple myeloma patients. Blood Coagul Fibrinolysis. 2007;18:595–98.
- Zangari M, Tricot G, Polavaram L, Zhan F, Finlayson A, Knight R, et al. Survival effect of venous thromboembolism in patients with multiple myeloma treated with lenalidomide and high‐dose dexamethasone. J Clin Oncol. 2010;28:132–35.
- L57 remain high, often over 10%, both in the clinical trial and real-world setting [17,18].
Answer: Would you like something changed/revised in this sentence? It is not clear from your comment.
- L60 why do you cite you here?
Answer: We have removed the citation from Fotiou et al, we cited the review as a summary of clinical trials but very happy to remove it. Thank you for your comment.
- L89 : specify the incidence which let you tell « VTE rates remain high also in the relapsed setting, as reported 89 recently in a meta-analysis.
Answer: Thank you for your comment we have changed the sentence in lines 108-109 to: “VTE rates remain high also in the relapsed setting, as reported recently in a meta-analysis; the VTE incidence per 100 patient-cycles was comparable in newly-diagnosed patients and in the relapsed setting at 1.2 and 1.2 respectively27”
- L93 : idem specify the lincidence « higher risk of developing VTE compared to the general population [1,2,15,16].
Answer: Thank you for your comment we have added the following to address your request: “In a large hospital-based study by Kristinsson et al14, during 17 years of follow-up, a 3.3-fold increase in DVT risk was observed in MGUS patients (n=2374) compared to the healthy population. More recently, another large population-based Danish cohort study reported an incidence rate of 4.0 VTEs⁄1000 person-years in MGUS patients and the IRR for VTE compared to a healthy control cohort matched with MGUS patients was 1.3728” see lines 114-119
- L96 : Could you explain the link between hypofibrinolysis and paraprotein? « Hyperviscosity, inhibition of natural anticoagulants, an increase of procoagulant inflammatory cytokines like interleukin-6 (IL-6) and tumor necrosis factor-a (TNFa) and hypofibrinolysis (due to monoclonal paraprotein interference) all contribute to the thrombotic risk»
Answer: We have changed the sentence to provide a possible explanation for the link between hypofibrinolysis and paraprotein. “There is data to support that hyperviscosity, inhibition of natural anticoagulants29, an increase of procoagulant inflammatory cytokines like interleukin-6 (IL-6) and tumor necrosis factor-a (TNFa) all contribute to the prothrombotic environment observed. Monoclonal paraprotein also interferes with fibrin polymerization31, clot retraction and clot lysis time 32, 33 and hypofibrinolysis is observed 30.”
- L 103 : all multiple myeloma treatment do not alter platelet function. Nowadays, anthracyclins are not used. It seems not justify this sentence « Altered platelet function has also been reported, which may explain the demonstrated efficacy of aspirin as thromboprophylaxis for low-risk MM patients »
Answer: Thank you for your comment. We have made the following revisions in the text: (Lines 132-136)“Altered platelet function has also been reported but data is not conclusive yet; Egan et al37 report decreased platelet aggregation in MM patients but no link between platelet hyporeactivity and paraprotein levels and Sullivvan et al report platelet hyperactivation with increased CD63, PAC-1 expression and annexin V in resting platelets of MM and MGUS patients compared to healthy controls38.”
- L109-112 : which reference ? « MM patients also had increased levels of cell-free DNA (cfDNA) as a surrogate of NETs. Further studies must determine whether these findings can be attributed to the procoagulant NETs activity. It should be noted that no patients in the study developed VTE, so a correlation between these findings and VTE occurrence cannot be demonstrated.
Answer: Thank you for pointing out the lack of reference, we have revised the text and have added the relevant references. Lines 141-146 “Cell-free DNA (cfDNA) is not a specific marker of NETs and increased thrombotic risk in MM patients but it may reflect NETs formation 42,44, 45. Recent data demonstrated that activation of PAD4 might induce NETs release directly in myeloma cell lines47. Further studies must determine whether these findings can be attributed to the procoagulant NETs activity. It should be noted that no patients in the study developed VTE, so a correlation between these findings and VTE occurrence cannot be demonstrated”
- L134 : Recent data are against. Please, add contradictory data : « The proteasome inhibitor (PI) bortezomib is associated with a low risk of VTE [50,51], 134 and even a protective effect has even been reported in some studies [47,52]. »
Answer: Thank you for your input. We have updated the sentence to read: (lines 181-184) “A protective effect has even been reported in some studies when bortezomib is combined with IMiDs 54, 59 but more recent data is conflicting; two RCTs did not show significant difference in the all-grade risk of thrombosis when IMiD plus dexamethasone is combined with bortezomib60, 61”
- L 174 Define « appropriate prophylaxis » («Only 19% of patients received appropriate prophylaxis[66]. ») (see lines 227-228)
Answer: “We have clarified the sentence and revised the sentence to: high risk and 18% low risk at baseline. Only 19% of patients received appropriate thromboprophylaxis (LMWH, aspirin, none) based on IMWG risk-score baseline assessment 76.”
- In the table : please add the definition « Dexamethasone (High-dose) (+4) Dexamethasone Low-Dose »for impede vte score.
Answer: We have added the dose in the table. Thank you
- What about gammaglobulines level data recently published?
Answer: We have found data on dysfibrinogenemia caused by immunoglobulin and have added this reference in lines 126-129: “Monoclonal paraprotein also interferes with fibrin polymerization30, clot retraction and clot lysis time 31, 32 and hypofibrinolysis is observed 33. More recently immunoglobulin k light chain was found to be associated with dysfibrinogenemia through a mechanism of fibrin interference34”
- L273 : Could you check the major hemorrhage incidence (1%) [81-84] in Storrar NPF, Mathur A, Johnson PRE, et al. Safety and efficacy of apixaban for routine thromboprophylaxis in myeloma patients 643 treated with thalidomide- and lenalidomide-containing regimens. Br J Haematol. 2019 Apr;185(1):142-144.
Answer: Thank you for your comment. We have rephrased the sentence to make it clearer that the rates reported is the pooled data from the 4 studies and not only the study by Storrar et al. “The safety and efficacy of apixaban thromboprophylaxis at 2.5 mg twice daily for at least six months in patients with NDMM who receive IMiDs have been evaluated in four recent trials in a total of 306 patients 92-95. In the pooled data two VTE events (0.6%) were reported, and three episodes of major hemorrhage (1%).”
- L 286 : platelet count >50000/mm3?
Answer: Thank you for pointing this out, we have corrected to <50 × 109/l
- L 296 : Are you sure ? « Drug-drug interactions and the use of strong cytochrome P inducers/ inhibitors should be taken into account; fortunately, no anti-myeloma agents (except dexamethasone) is known to be a potent inhibitor or inducer of P-glycoprotein and P450 [86]. » See reference Sorigue 2020.
Answer: Thank you very much for pointing this out. We have revised the sentence (line 383-385). “Drug-drug interactions and the use of strong cytochrome P inducers/ inhibitors should be taken into account. Among anti-myeloma agents, dexamethasone is a strong inducer of CYP3A4 and P-glycoprotein and inhibitor of P-glycoprotein, bortezomib is a CYP3A4 inhibitor and venetoclax an inhibitor of P-glycoprotein97, 98”
- You referred NCCN 2022 as 10 instead of 69. NCCN Clinical Practice Guidelines in Oncology (NCCN Guidelines): Cancer-Associated Venous Thromboembolic Disease.: 487 National Comprehensive Cancer Network; July 22, 2016. [updated Version 1. 2016; cited 2018 December 12]. Available from: 488.
Answer: Thank you we have revised the reference.

Reviewer 2 Report
This is an informative review which addresses an important unmet need in MM. Although there are now several published reviews on this topic this review addresses the risk factors for VTE very well and also summarises the data on arterial thrombosis which is lacking in published reviews.
Major comments;
· The main strength of this review is the comprehensive presentation of the published literature that relates to risk assessment models, and biomarkers, of VTE risk in MM. The review addresses the literature of risk factors very elegantly but also the prevention and treatment recommendations. Therefore, I would suggest to amend the title of the review from “management and prevention” to “risk assessment, prevention and management”.
· The authors mention in 2.1.3 the different risks associated with the use of IMIDS, mAbs etc and some of the post hoc analysis of GRIFFIN etc. It would be useful if the authors could present a table of the VTE risk associated with individual agents e.g. Len, pom, dara etc. This could potentially replace Table 1, if necessary, which is not too novel.
· Section 2.4 and 2.4.1 can be reduced in length and combined. The guideline recommendations can be summarised and referenced within the same paragraph as the data around suggested agents. As it stands these two sections do not make sense to have apart as it is a little repetitive.
· Discussion can be shortened – e.g. the point about clinical trials is also made in the conclusion so does not need to be in the discussion.
Minor comments;
· Line 33-36 sentence beginning with “The emergence of VTE” – please revise grammar
· Line 38 “triple the mortality rate” – this is not well established in the referenced studies, while the increase in mortality is increased in the studies it is difficult to compare hazard ratios between studies. For example in the referenced Kristinsson study the hazard ratio for mortality was slightly lower than triple. I would suggest to revise this sentence.
· Line 66 “ the shortcomings of current guidelines” – this sentence should be removed.
· Again for reading I would suggest to revise sentence beginning with “advanced age” lines 74-77 – too long.
· Please revise this sentence line 88 “Newly diagnosed MM disease is considered a risk factor compared to relapsed/refractory disease, however, “
· Line 101 - in relation to the role of VWF in thrombosis mechanisms in myeloma a recent review (PMID 35644028) high plasma VWF levels have been shown to increase VTE risk and reduce overall survival. Authors could include this recent review as a reference.
· Line 125 “Data for pomalidomide come from the relapsed setting and following the implementation of regular thromboprophylaxis, particularly in the clinical trial context [44-46].” Could the authors expand on this? The review does not make it clear here if pomalidomide increases risk, if this is in the trial setting or outside of the trial setting. The data for pomalidomide should be more clearly outlined.
· Line 329 sentence starting with “the authors” – does this refer to Stefano et al or the authors of the review? Please clarify this point.
Author Response
Please sed attachment
This is an informative review which addresses an important unmet need in MM. Although there are now several published reviews on this topic this review addresses the risk factors for VTE very well and also summarises the data on arterial thrombosis which is lacking in published reviews.
Major comments;
- The main strength of this review is the comprehensive presentation of the published literature that relates to risk assessment models, and biomarkers, of VTE risk in MM. The review addresses the literature of risk factors very elegantly but also the prevention and treatment recommendations. Therefore, I would suggest to amend the title of the review from “management and prevention” to “risk assessment, prevention and management”.
Answer: Thank you for your comments. We have revised the title accordingly
- The authors mention in 2.1.3 the different risks associated with the use of IMIDS, mAbs etc and some of the post hoc analysis of GRIFFIN etc. It would be useful if the authors could present a table of the VTE risk associated with individual agents e.g. Len, pom, dara etc. This could potentially replace Table 1, if necessary, which is not too novel.
Answer: Thank you very much for your input. We have added a table (table 1) with the VTE risk associated with individual agents in the novel era of treatment regimens. We have not removed table 1 (now 2) as it is not novel but summarizes the risk scores nicely.
- Section 2.4 and 2.4.1 can be reduced in length and combined. The guideline recommendations can be summarised and referenced within the same paragraph as the data around suggested agents. As it stands these two sections do not make sense to have apart as it is a little repetitive.
Answer: We have merged and shortened sections 2.3 and 2.4 as requested to avoid repetition. Thank you for your comment.
- Discussion can be shortened – e.g. the point about clinical trials is also made in the conclusion so does not need to be in the discussion.
Answer: Thank you for your input, we have revised the discussion and shortened it and we have removed the point about clinical trials from the discussion.
Minor comments;
- Line 33-36 sentence beginning with “The emergence of VTE” – please revise grammar.
Answer: We have revised the sentence to read: “The emergence of VTE as a significant complication in MM patients parallels the use of multiagent chemotherapy regimens (doxorubicin, vincristine and more) and later on the introduction of immunomodulatory agents (IMiDs). The association between IMiDs and VTE risk is well established in the literature3, 4.”
- Line 38 “triple the mortality rate” – this is not well established in the referenced studies, while the increase in mortality is increased in the studies it is difficult to compare hazard ratios between studies. For example in the referenced Kristinsson study the hazard ratio for mortality was slightly lower than triple. I would suggest to revise this sentence.
Answer: Thank you for your comment we have revised the sentence to: “The association between IMiDs and VTE risk is well established in the literature3, 4. Both venous and arterial thrombosis are a leading cause of morbidity but evidence of the association between VTE and mortality in MM is limited and conflicting.5, 6, 7,8”
- Line 66 “ the shortcomings of current guidelines” – this sentence should be removed.
Answer: “The sentence has been revised to: In this review, we present current approaches to management and prevention of thrombotic complications in patients with MM and we highlight the necessity better to characterize the unique coagulation profile of the MM patient.”
- Again for reading I would suggest to revise sentence beginning with “advanced age” lines 74-77 – too long.
Answer: We have revised the sentence to: “Standard risk factors for thrombotic risk apply for the MM patient22,23”
- Please revise this sentence line 88 “Newly diagnosed MM disease is considered a risk factor compared to relapsed/refractory disease, however, “ Thank you for your comment. We have revised the sentence to read:
Answer: “A new diagnosis of multiple myeloma is considered a risk factor for thrombosis compared to relapsed/refractory disease4, however, VTE rates remain high also in the relapsed setting, as reported recently in a meta-analysis; the VTE incidence per 100 patient-cycles was comparable in newly-diagnosed patients and in the relapsed setting at 1.2 and 1.2 respectively26.”
- Line 101 - in relation to the role of VWF in thrombosis mechanisms in myeloma a recent review (PMID 35644028) high plasma VWF levels have been shown to increase VTE risk and reduce overall survival. Authors could include this recent review as a reference.
Answer: Thank you for pointing out this review, we have included it in the reference list. “A recent review summarizes data regarding the association between increased vWF/FVIII levels in patients with MM, increased risk of thrombosis and decreased overall survival 37”
- Line 125 “Data for pomalidomide come from the relapsed setting and following the implementation of regular thromboprophylaxis, particularly in the clinical trial context [44-46].” Could the authors expand on this? The review does not make it clear here if pomalidomide increases risk, if this is in the trial setting or outside of the trial setting. The data for pomalidomide should be more clearly outlined.
Answer: Thank you for your comment. We have revised this part of the paper to make clearer the risk associated with pomalidomide. “Fewer data exist regarding pomalidomide; the VTE risk reported is around 2-3% when pomalidomide is administered in combination with dexamethasone. The VTE risk is therefore lower compared to lenalidomide-based combinations. It should be however taken into account that data come from the relapsed setting only and following the implementation of regular thromboprophylaxis in this patient population52-54. “
- Line 329 sentence starting with “the authors” – does this refer to Stefano et al or the authors of the review? Please clarify this point.
Answer: We have clarified that we refer to Stefano et al and the consensus paper. “According to consensus paper, prophylactic dose LMWH should be the first choice for all other patients. Stefano et al.”

Round 2
Reviewer 1 Report
For question 13 :
Prediction of venous thromboembolism in patients with multiple myeloma treated with lenalidomide, bortezomib, dexamethasone, and transplantation: Lessons from the substudy of IFM/DFCI 2009 cohort. Chalayer el al. JTH
"serum gamma globulin level > 27 g/L (2.8 [1.2–6.8,] P = .02)."
Author Response
Question 13:
Thank you very much for providing the reference.
Reviewer 2 Report
Now that this manuscript has been appropriately revised and is acceptable for publication
Author Response
thank you